# Use of Level-of-Service-Based Water Pricing to Sustain Irrigated Agriculture: A Case of Modernized Irrigation System in Vietnam

**Truong Duc Toan** *[ID] **and Bui Anh Tu** [ID]

Department of Economics, Faculty of Economics and Management, Thuyloi University, Dong Da District, Hanoi 100000, Vietnam
* Correspondence: toantd@tlu.edu.vn

**Abstract:** Water is an important input for irrigated agriculture. However, the irrigation sector, especially in developing countries, often faces pressure to secure water for production activities and maintain irrigation services. To sustain the supply and delivery of water in irrigation, not only should efficient use of water resources be promoted, but the balance between the cost and revenue from water supply must be sustained. In most cases, the appropriate setting and application of water pricing is crucial to achieving these objectives. In this paper, the use of level-of-service-based water pricing is described and illustrated with a case of a modernized irrigation system in a central highland province of Vietnam. The results from this study show that: (i) modernization of irrigation systems increases service levels and farmers have more choices for selecting services provided; (ii) water rates can be reasonably calculated with respect to the level of irrigation services provided; and (iii) farmers are more likely to select an acceptable level of service for a reasonable water price instead of choosing to pay more for a higher quality of service. This paper highlights the importance of policies which support the modernization of irrigation systems and pursuing level-of-service-based water pricing to promote more sustainable irrigation development in developing countries.

**Keywords:** water pricing; level of service; cost reflective pricing; irrigation policy; sustainable irrigation development

## 1. Introduction

Irrigation is an important input for agricultural production. However, in the context of climate change, many countries face pressures in dealing with the fluctuation in water quantity, as well as insecure access to reliable irrigation services [1]. Efficient water use and financial stability of irrigation services are crucial for sustainable irrigation development in this context. The reliability of irrigation services can be improved through solutions such as upgrading or modernizing irrigation systems to secure the water supply. However, such systems will only be sustainable if they are accompanied by economic and governance tools, which enhance the efficient use of water resources in agricultural production and generate sufficient finances to sustain irrigation services. In these cases, water pricing is considered an appropriate option to pursue. Water pricing, in general, can be used to serve a number of objectives [1] (p. 60):

i.   Optimizing the allocation of water among competing uses;
ii.  Sustaining a balance between the expenditure and revenue from the water supply to cover investment, operation and maintenance;
iii. Managing demand and discouraging the waste of water resources;
iv.  Offering appropriate prices as well as adequate and equitable access to affordable water and water-related services.

In the provision of irrigation services, two main purposes of water pricing are important: (i) to secure the financial sustainability of water services and water suppliers; and (ii) to increase water use efficiency. From the water supplier's perspective, cost recovery

is essential because if the cost is not fully covered, it does not guarantee an effective and efficient operation and management of the water system, or sufficient investment in water infrastructure. Nevertheless, the question of how to set a water price correctly remains challenging due to many influencing factors regarding the complexity of the hydro-economic system and the response of water demand to price changes [2]. In this respect, different pricing strategies can be applied to recover supply costs even in cases with the absence of water metering [3]. Meanwhile, from the perspective of the water user, the application of variable water prices linked to consumption disincentivizes wasteful usage of the resource because the marginal cost will surpass the marginal benefit of each unit of water consumed [4]. Water users often respond positively to the increase of water prices, and the amount of water used for irrigation can be reduced considerably in response to water pricing [5,6], including in the case of water conservation [7].

Irrigation water pricing has been widely discussed in the literature but it is still a challenge for the scientific community [8]. A number of irrigation water pricing methods that are commonly applied consist of: area-based pricing (a fixed rate or area–crop-based charge); a volumetric charge (supposed to represent the marginal costs of water provision); block pricing (tiered pricing); a two-part tariff; property rights and water markets. Great efforts at a global scale have been made to share experiences from various countries or to compare the efficiency of different pricing methods. For example, see Dinar and Subramanian [9], Johansson, et al. [10], Molle and Berkoff [11], Dinar, et al. [12] and Toan [13]. However, the importance of considering level-of-service (LoS) in water pricing methods is still neglected in the literature. In fact, very few studies focus on the necessity of taking into account service quality, by specifying the factors affecting service level, to improve the management and operation of irrigation systems.

One exception is Malano and Van Hofwegen [14], who highlight the need for a service orientation in the management of the irrigation sector. In this case, the authors argue that the level of service should be specified in terms of adequacy, reliability, equity and flexibility. Malano and Van Hofwegen [14] also emphasize the requirement of specifying service levels for both new and rehabilitated systems, stating that irrigation systems need to be managed as a service business responsive to the needs and changing requirements of its customers. This highlights two key determinants in the success of sustainable irrigation systems: transparency of the costs associated with maintaining high LoS, and the opportunity for water users to balance LoS against the ability to pay for the service.

More studies have shown the efficiency of applying the LoS concept in different areas such as Shurtz, et al. [15], in exploring water use for irrigation in urban settings and the role of pricing in encouraging users to match their water use to the LoS, or Atwater, et al. [16], in suggesting an individualized approach to determining water rates based on customers' water-use patterns using smart meters, and Han, et al. [17], in recommending the use of customer-driven LoS to set appropriate service levels in water infrastructure asset management. In Australia, rural water prices are determined based on the cost of supply as "cost reflective pricing" and crucially reflecting the LoS provided [18]. This aimed to enable change or the optimization of behavior of both water users and the system operator. The benefit of the introduction of the two-way "reflective" approach is that it puts pressure on both buyer and seller to find the right price for the level of service. There is:

- Pressure on the delivery system operator to:
  - Provide the "right" LoS because the farmers will not want to pay for an inadequate LoS;
  - Be efficient in their operations and to keep costs to a minimum because the farmers do not want to pay more than absolutely necessary.

- Pressure on the farmer to:
  - Only demand a LoS that they are prepared to pay for, otherwise the costs will be excessive;
  - Make economic decisions about how they use water—this may mean they are more efficient in water use, but not necessarily.

Thus, efficient and cost reflective water pricing would lead to an appropriate LoS provided to the farmer at an affordable cost. There is an opportunity to consider the application of this experience more widely, and in the contemporary context of climate and development pressures.

The literature in the field of irrigation also shows the important role of modernizing irrigation systems and applying volumetric water pricing to them to sustain water services. A number of scholars assert that the modernization of irrigation systems aims to enhance irrigation efficiency and to achieve other positive outcomes (for example, see [8,19]). Parween, et al. [20] insist that volumetric pricing needs to be adopted to increase the efficiency of water use and management. Chaudhuri and Roy [21], in reviewing the irrigation water pricing policy in India, advise applying a volumetric water pricing system and installing automatic meters to charge for the actual volume of water used. However, policy makers and practitioners must also take into account other issues that may emerge in regard to the sustainability of supplying irrigation services and the water user's willingness to pay. Several studies emphasize the impact on the environment when a volumetric charging system with higher water prices is applied, since farmers may shift the form of water usage from surface water to groundwater [22,23]. Nevertheless, water pricing with volumetric charging is highly recommended to incentivize more efficient water use in the literature.

In the case of Vietnam, this country is known as one of the countries most affected by climate change [24]. In this context, national water security has been a topic of discussion in many discourses, even at the highest level of governance. The topic of ensuring national water security has been proposed to the Political Bureau of the Party Central Committee and the Bureau has issued the Politburo Conclusion 36-KL/TW, dated 23 June 2022 on ensuring national water security, reservoir and dam safety until 2030, with a vision to extending this til 2045. The conclusion provides important directions for the water development of Vietnam in the coming years. A strategy is being prepared to pursue new approaches toward more sustainable development of the water sector in the future.

Over the last three decades, Vietnam has applied different approaches toward water pricing, and the concept of irrigation fees and water fees has been common. When the Law on Hydraulic Works was enacted and implemented from July 2018, the concept of water prices for irrigation products and services was applied on a national scale. However, the flat rate method of setting prices for irrigation services is normally used (VND/ha). This approach does not incentivize either the maintenance of reliable irrigation services or efficient water use, as it does not account for the LoS or volume of water supplied. Nevertheless, Vietnam is moving toward more sustainable water resource management by implementing a nationwide water pricing reform as indicated by the Law on Hydraulic Works. The ongoing reform that has recently been focused on is the implementation of irrigation water pricing mechanisms. Different pricing methods are more likely to be promoted in the current context, since new technologies such as remote sensing and control, cloud computing and decision support systems are available to help to change methods of water use and irrigation management in the agricultural sector. Nevertheless, the challenges of irrigation water pricing in practice are numerous and it is not easy to deploy effectively [25]. For example, the requirements of an adequate legal framework, institutional and administrative resources capable of implementing and enforcing the policy and the acceptance of the end user are crucial.

The aim of this paper is to present the LoS-based water pricing method as a viable method promoting sustainable irrigation development. This approach is also based on the pricing theory that considers demand management as a central point from which to promote efficient use of water and to recover the cost of water supply. However, compared to the common approaches, the current method involves a process that ties the relationship between the provider and water users together in decision making: to select a desired LoS and then to accept a determined price. A mechanism that involves the participation of both sides in decision making is more likely to lead to a better result. This paper illustrates this method with the case of a modernized irrigation system supplying water for the high value

crops of an agricultural cultivation area in a central highland province of Vietnam. Cost reflective pricing of the water supply and delivery is considered with respect to assumed LoS options. This paper also describes how a change in the LoS would affect the cost of the water supply, and then the price of irrigation services, through simulations of the hydraulic performance of the system to estimate the cost of a reduced LoS and a survey of farmer acceptance of the new prices. This study highlights the importance of policies that modernize irrigation systems and pursue LoS-based water pricing in promoting sustainable irrigation development in various countries.

## 2. Methodology

### 2.1. LoS-Based Water Pricing

LoS-based water pricing is the method which determines water price based on each defined LoS for the irrigation system. When water price is determined by this approach, it is critical to clearly identify and clarify different LoS tiers because the cost of water supply will differ in response to each LoS. Identifying and clarifying different LoS tiers also provides transparency to water users. Each LoS can be assumed regarding various factors. A number of factors can be taken into account: volume of water supplied, time of supply, reliability of the water supply, equity among water users and cost.

- **Volume:** the seasonal supply volume (the quantity, seasonal variability and water quality).
- **Offtake:** delivery or service point conditions (channel capacity, off-take flow rate, off-take elevation/pressure).
- **Scheduling:** relative flexibility of supply (continuous flow, rotation, on order or on demand).
- **Reliability:** control and operation of structures to supply the intended quantity and flow consistently.
- **Equity:** ensuring water supply to the lower end of canals.
- **Cost:** the costs for current service delivery.

In respect to a certain LoS provided, it is possible to determine the costs of irrigation services and then the price of irrigation water. The price reflects a change in the cost of water supplied. Options with different LoS tiers will be evaluated in order to identify the optimum operating and water pricing conditions suitable for each system. With modernized irrigation systems, water price is usually calculated based on the volumetric unit of the water quantity supplied. This cost will be influenced by cost elements and can change over time. The cost of each quantity of water supplied can be calculated for the cost of each unit of incremental water supply, or based on the total water supply to the water users.

Water price can be calculated using one of the two methods: (1) average cost pricing method; and (2) two-component cost pricing. This will depend on the decision made based on the preference of the service users. The two-component water pricing method is considered for the variety of choice it offers to water providers and farmers.

1. Average cost pricing method:

The average cost pricing method is calculated as total cost divided by the quantity of water supplied. The total cost consists of direct material cost, direct labor cost, general production cost, management cost, sales cost, expected profit, VAT (if any). The formula to calculate water price is given below:

$$P_{AVC} = \frac{TC}{Q} \tag{1}$$

where:

- $P_{AVC}$: water price calculated using average cost method (VND/m$^3$).
- $TC$: total cost of water supply and delivery (VND/year).

- 　　*Q*: total quantity of water supplied (m³/year).

2. 　Two-component cost pricing:

　　In this case, the price of irrigation water is structured by two components: fixed cost and variable cost components. Fixed cost is considered as the cost that does not change with respect to a change in the quantity of water supplied or used. The fixed cost component consists of the investment cost of the irrigation system, the general cost for management and operation of the system. Variable cost is the cost component that varies with respect to a change in water quantity supplied. Variable cost could consist of electricity cost for pumping water or other costs. The formula to calculate water prices is given below:

$$P_{FC} = \frac{TFC}{TIA} \tag{2}$$

$$P_{VC} = \frac{TVC}{Q} \tag{3}$$

where:

- 　*P_{FC}*: price of irrigation water caculated for fixed cost component (VND/year/ha).
- 　*TFC*: total fixed cost (VND/year).
- 　*TIA*: total irrigated area (ha/year).
- 　*P_{VC}*: price of irrigation water calculated for variable cost component (VND/year).
- 　*TVC*: total variable cost (VND/year).
- 　*Q*: total quantity of water supplied (m³/year).

　　The advantage of the two-component water pricing method is that it helps to determine economically efficient prices where the calculation of water price is based on variable costs. These cost elements often vary depending on the LoS of irrigation. Fixed costs are considered as an annual amount paid by water users. These costs may change if the irrigation system is invested in further for expansion or modernization.

　　In the current study, cost components are structured as follows:

- Direct material cost: consisting of the energy cost for operating the pump irrigation systems. This energy cost is calculated using the formula below [26]:

$$G_{\text{electricity}} = \frac{\gamma \times Q_p \times H_p \times T \times 365 \times g_e}{102 \times 3.6 \times \eta_{pp} \times \eta_{mp} \times 10^6} (\text{US\$}) \tag{4}$$

where:

- 　$G_{\text{electricity}}$: cost of electricity per year (US\$/year).
- 　$\gamma$: density of water (kg/m³).
- 　$Q_p$: total flow of the pumping station (m³/h).
- 　$H_p$: total head of the pump (m).
- 　*T*: number of hours of pumping (h).
- 　$g_e$: electricity price per kW (US\$/kW).
- 　$\eta_{mp}$: electric motor performance (%).
- 　$\eta_{pp}$: pump performance (%).

- Direct labour cost: comprising direct labor cost (in the current case study this includes salaries for two persons working full time, salary-based payments including social insurance, health insurance, unemployment insurance, union fees, labor safety, uniforms and other costs such as responsibility allowance, meal allowance, motorbike fuel and telephone allowances).
- General production costs: consisting of depreciation cost of fixed assets, machine maintenance cost, connection maintenance cost, tax/fees on water rights (if any).
- Sales costs: comprising salary of salesmen (including salary for one person working full time, salary-based payments, allowances, stationary cost and water bill printings.

- Management cost: consisting of salaries for administrative, manager, guarding and cleaning labor (with salary-based payments, allowances, cost of office tools and assets, maintenance and repairs, cost of equipment, training cost, hospitality cost, conferences, asset insurance and other costs. In the current case, management cost for irrigation activities is estimated at 30% of the total management cost of the cooperative.
- Expected profit: calculated by 5% of total production cost as regulated by the government in the area of irrigation service provision.
- VAT tax (if any).

Once the LoS and water price are identified with the consent of the supplier and the water users, the supply and use of irrigation services is deployed in practice. The process and procedure of the supply and use of irrigation services involves five steps as follows:

a. Preparing a plan for the supply and use of irrigation services in accordance with water users' requirements before each irrigation season or irrigation period;

b. Determining water price frame/options that reflect the assumed LoS and appropriate cost elements;

c. Undertaking negotiations and signing contracts between the service provider and the farmers, taking into account irrigation conditions, water demand and supply capacity;

d. Conducting the provision and use of the services in accordance with the given contracts;

e. Undertaking the acceptance of completed services and liquidating the irrigation service contracts by each irrigation season.

*2.2. Case Study*

The LoS-based water pricing method is applied to the case of the pumping irrigation system of Thanh Nghia Agriculture Cooperative (AC) in Don Duong district, Lam Dong Province of Vietnam. The cooperative was formed in 1978 and currently has 233 members. The study location is shown in Figure 1.

The irrigation system of the cooperative provides an irrigation service with a total agricultural production area of 102 ha. The area is intensively irrigated, with high value vegetable and flower crops. The irrigation water was originally supplied from the river by a pumping station, which lifted the water into low level canals that worked as both supply and drainage canals. The farmers used their own individual low lift pumps to supply water to piped farm irrigation systems (sprinklers and dripping). During the dry seasons of 2014–16 there was a shortage of water and this, coupled with aging pumping system, caused high risk in the water supply. This resulted in a new modern irrigation system being proposed and built.

The new pumping station consists of 3 Tsurumi submersible pumps alternately operating in inverter mode, 1 mud pump and 1 inverter control cabinet with technical indicators as below:

- One submersible pump with capacity of 57 kW; Q = 241 $m^3$/h; Hmax = 80 m;
- Two submersible pumps with capacity of 90 kW; Q = 290 $m^3$/h; Hmax = 80 m;
- One mud pump with capacity of 2.2 kW; Q = 60 $m^3$/h; H = 22 m.

The inverter control cabinet will automatically control the capacity of the system according to actual irrigation through one electronic flow meter D600 and four pressure sensor points on the pipeline system. The HDPE pipeline system has 10 lines of pipes with a total length of 5919.47 m. There are 5 positions of sludge discharge valves, 2 positions of air discharge valves, 78 locations for water intake with 3 holes of meter valves per position to distribute water to the farms. The technical parameters of the pipeline system are given in Table 1.

The layout of the pipeline network of the irrigation system is given in Figure 2.

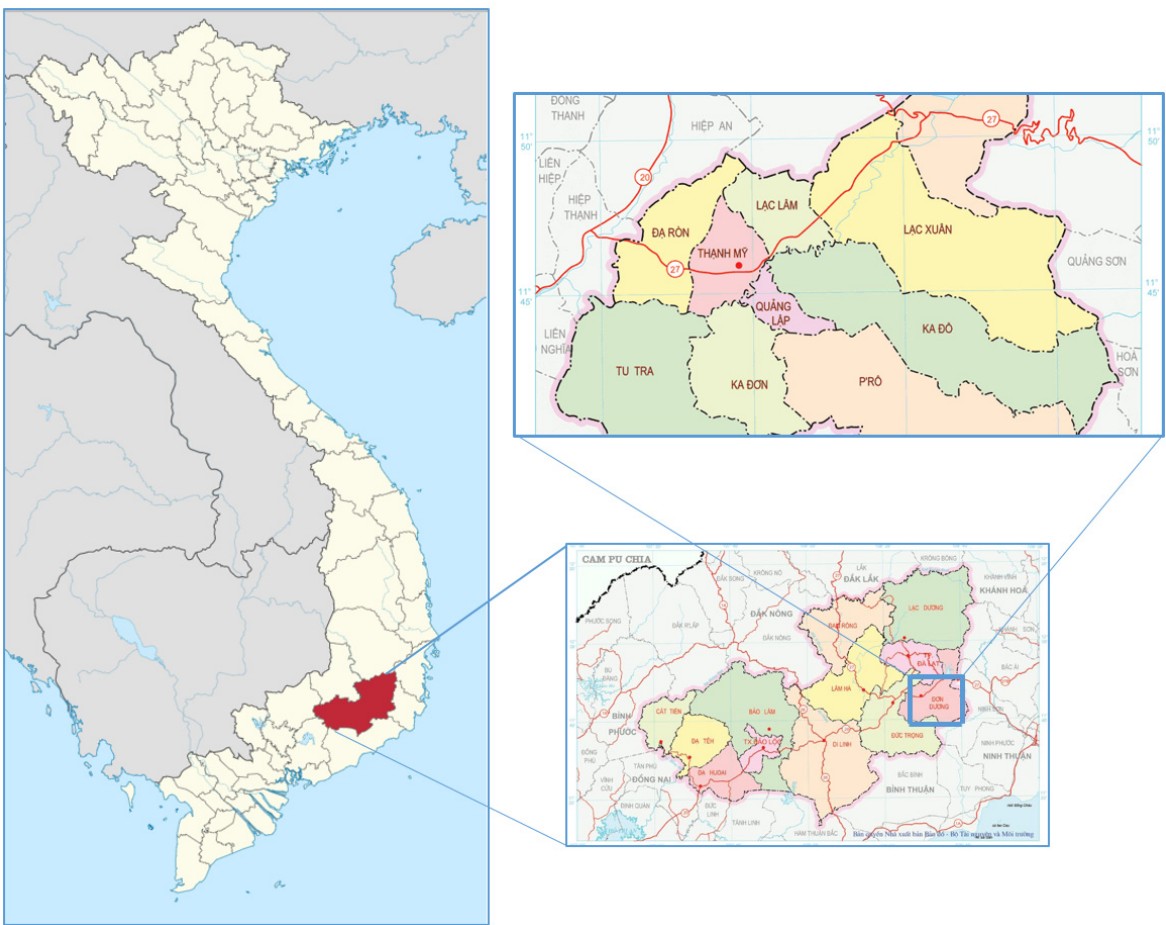

**Figure 1.** Location of the case study.

**Table 1.** Technical parameters of the pipeline system.

| No | Name of Pipeline | Flow of Pipeline Head (L/S) | Irrigated Area (ha) | Length of Pipeline (m) | Type of Pipeline | Diameter of Pipeline (mm) | Number of Valve Pits |
|---|---|---|---|---|---|---|---|
| 1 | Line OC | 221.85 | 102 | 23.82 | T | 2D315 | |
| 2 | Line N1 | 73.08 | 8 | 1042.92 | HDPE | 315 | 14 |
| 3 | Line N1-1 | 55.68 | 7.6 | 287.25 | T + HDPE | 250 | 11 |
| 4 | Line N1-3 | 39.15 | 18 | 1014.4 | HDPE | 200 | 9 |
| 5 | Line N2 | 148.77 | 29.6 | 1233.6 | HDPE | 315 | 18 |
| 6 | Line N2-1 | 14.36 | 6.6 | 544.9 | HDPE | 200 | 8 |
| 7 | Line N2-2 | 21.75 | 10 | 587.52 | HDPE | 200 | 8 |
| 8 | Line N2-3 | 11.96 | 5.5 | 450.67 | HDPE | 200 | 4 |
| 9 | Line N2-5 | 9.57 | 4.4 | 346.79 | HDPE | 200 | 4 |
| 10 | Line N2-7 | 26.75 | 12.3 | 387.6 | HDPE | 200 | 6 |
| **Total** | | | **102** | **5919.47** | | | **82** |

Specification of the LoS of Thanh Nghia Irrigation System

In Thanh Nghia irrigation system, a number of LoS elements were specified as below:

- **Volume:** there are no water constraints regarding the irrigation system. Water is taken directly from Da Nhim river, and the water source from this river is available over the year. Experience from the irrigator shows that even in years with droughts, water volume is still assured for the operation of the pumping station. Therefore, this element will not be considered as a variable factor affecting the water price of the current case.
- **Offtake:** Thanh Nghia irrigation system has direct pipe connections, therefore, it can deliver its service to every farm individually. The pump system can be operated at

different pressures of 40–80 m head (see the hydraulic calculations in Section 3.2). In addition, there is no need for the farmer to pump. However, an insufficient flow rate or short time of water supply may affect the LoS for some households in this case. This element will be taken into account in the water price calculation.

- **Scheduling:** different time frames of water supply may affect how farmers irrigate. Serving water on demand will increase the cost of service provision. The current irrigation system was designed with appropriate pipe sizes and farmer outlets, therefore, every farmer can irrigate. The length of the irrigation season does not affect the system operation. However, different pumping times within a day will influence the cost of the water supply, since the electricity charge varies with various time-frames in Vietnam (see Section 3.3). This will be examined for the impact of the time factor on the water price.
- **Reliability:** the supply of water with consistency such as keeping pressure throughout the system, provision of water with intended quantity, availability of irrigation service in all seasons. With the current pressurized irrigation system, this factor does not, therefore, affect the water price.
- **Equity:** Every farmer gets a similar LoS throughout the whole system and farmers are charged equally via measurement meters and price mechanisms. This factor in the current system does not cause any changes to the water price and, therefore, it is not be considered in the price setting.

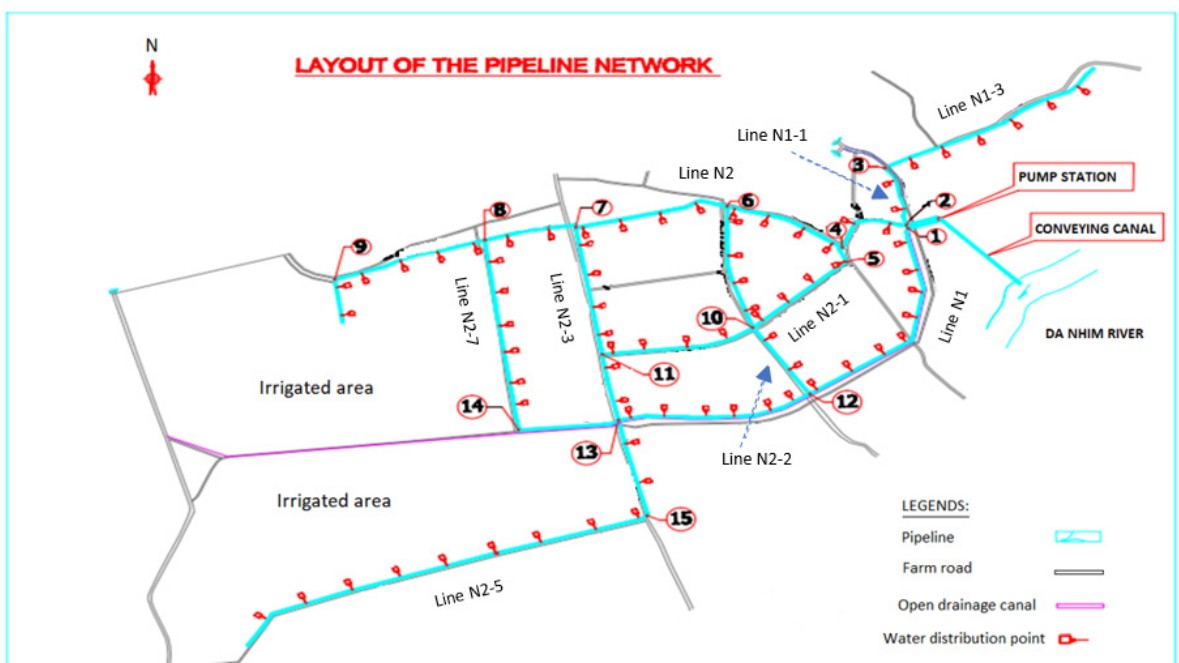

**Figure 2.** Layout of the pipeline network of Thanh Nghia irrigation system (Numbers 1–15 are of water intake positions).

From the given data on the pumping irrigation system, a new water pricing method was proposed to apply to the system. The objective of water pricing is to increase farmers' income through sustainable production, to ensure a stable water supply for agricultural production and to protect the environment through preventing water leakage and soil erosion from the water conveyance system. The method of average cost is applied to determine the water price for the case of Thanh Nghia pumping irrigation system and the water supplier is fully paid for all costs. The price of irrigation service is calculated from the total cost of water supply and delivery divided by the quantity of water supplied to the pumping station system. The total cost is calculated based on all necessary expenses for the operation and management of the pumping station from the headwork to the household's

meters. The calculation was undertaken for different water prices in respect to assumed LoS options.

## 3. Results

### 3.1. Hydraulic Considerations

Hydraulic calculations with different operation options were undertaken to check the ability of the system to supply water to the irrigated area. The purpose of these calculations is, firstly, to determine the LoS provided to the individual farmer and how it varies across the system and with varying flow rates; secondly, to determine the energy required to deliver the water and, thirdly, to discover what would be the impact of reducing the delivery pressure. A key purpose of hydraulic calculations is to determine the energy required per unit of water delivered ($kWh/m^3$), as energy costs (electricity) in a pressurized pipeline system occupy a major part of the system cost.

The EPANET program has been used to undertake hydraulic calculations for Thanh Nghia irrigation pumping system. This model is commonly used to calculate hydraulics of pumping systems to recommend efficient solutions for operation of the network [27]. Water demands withdrawn at specific nodes are determined based on the type of crops and the area that each node will serve. The application of this model requires the user with experience in operating irrigation systems to draw appropriate simulation scenarios. Generally, a delivery pressure of 30 m would be sufficient for water delivery at the farm level. Therefore, hydraulic calculations are made to find out the most suitable delivery pressures for the pumping system. As the design capacity, the pressures at junctions with an operating head of 80 m were calculated and given in Table 2.

**Table 2.** Pressure at junctions in case of an operating head of 80 m.

| Node ID | Elevation (m) | Demand (LPS) | Head (m) | Pressure (m) |
|---|---|---|---|---|
| Junc J-1 | 12.59 | 14.52 | 86.15 | 73.56 |
| Junc J-2 | 12.3 | 11.92 | 85.74 | 73.44 |
| Junc J-3 | 11.59 | 14.79 | 85.13 | 73.54 |
| Junc J-4 | 11.59 | 13.82 | 84.75 | 73.16 |
| Junc J-5 | 11.25 | 18.49 | 84.64 | 73.39 |
| Junc J-6 | 10.29 | 7.23 | 84.63 | 74.34 |
| Junc J-7 | 10.34 | 13.91 | 85.10 | 74.76 |
| Junc J-8 | 11.07 | 13.55 | 84.74 | 73.67 |
| Junc J-9 | 12.75 | 19.49 | 85.12 | 72.37 |
| Junc J-10 | 12.59 | 17.10 | 84.73 | 72.14 |
| Junc J-11 | 12.39 | 11.54 | 84.63 | 72.24 |
| Junc J-12 | 11.29 | 9.53 | 84.26 | 72.97 |
| Junc J-13 | 11.09 | 15.04 | 83.92 | 72.83 |
| Junc J-14 | 9.21 | 9.66 | 83.81 | 74.60 |
| Junc J-15 | 13.21 | 10.89 | 85.87 | 72.66 |
| Junc J-16 | 12.35 | 8.53 | 85.65 | 73.30 |
| Resvr 1 | 6.3 | −210.00 | 6.30 | 0.00 |

The calculations show that, in the current situation, the pumping system capacity by design is much greater than that of requirement. Furthermore, it is apparent that the pressure provided is greater than is required to meet the farmers' needs. A number of scenarios of pumping operation were calculated with operating heads of 70 m, 60 m, 50 m and 40 m to check the ability of the system to supply water to the farmer. The calculation results with an operating head of 40 m are shown in Table 3.

The results from the table above indicate that with an operating head of 40 m, the pressure would be reasonable to supply water to individual farms.

In addition, three design operating conditions were also considered:

i.　Maximum flow of 210 L/s with all 82 hydrants delivering 5 L/s for a given pressure (i.e., 80 m head);

ii.　Zero flow where pressure around the whole system equals the control point pressure (i.e., 80 m);

iii.　Reducing the delivery pressure to 30 m at the control point, which requires a pumping head of approximately 40 m.

From this, the following can be determined:

a.　The range in pressures for the individual hydrants across the system under full flow conditions;

b.　The range in pressures for each individual hydrant from full flow to zero flow conditions;

c.　The hydraulic pumping head required for full flow conditions and for zero flow conditions;

d.　The energy required (kWh) per $m^3$ (which can be determined from the maximum flow operating conditions);

e.　The reduction in energy ($kWh/m^3$) from operating at the lower pressure.

**Table 3.** Pressure at junctions in the case of an operating head of 40 m.

| Node ID | Elevation (m) | Demand (LPS) | Head (m) | Pressure (m) |
|---|---|---|---|---|
| Junc J-1 | 12.59 | 14.52 | 46.23 | 33.64 |
| Junc J-2 | 12.3 | 11.92 | 45.82 | 33.52 |
| Junc J-3 | 11.59 | 14.79 | 45.21 | 33.62 |
| Junc J-4 | 11.59 | 13.82 | 44.82 | 33.23 |
| Junc J-5 | 11.25 | 18.49 | 44.72 | 33.47 |
| Junc J-6 | 10.29 | 7.23 | 44.70 | 34.41 |
| Junc J-7 | 10.34 | 13.91 | 45.18 | 34.84 |
| Junc J-8 | 11.07 | 13.55 | 44.81 | 33.74 |
| Junc J-9 | 12.75 | 19.49 | 45.20 | 32.45 |
| Junc J-10 | 12.59 | 17.10 | 44.80 | 32.21 |
| Junc J-11 | 12.39 | 11.54 | 44.70 | 32.31 |
| Junc J-12 | 11.29 | 9.53 | 44.33 | 33.04 |
| Junc J-13 | 11.09 | 15.04 | 43.99 | 32.90 |
| Junc J-14 | 9.21 | 9.66 | 43.89 | 34.68 |
| Junc J-15 | 13.21 | 10.89 | 45.95 | 32.74 |
| Junc J-16 | 12.35 | 8.53 | 45.73 | 33.38 |
| Resvr 1 | 6.3 | −210.00 | 6.30 | 0.00 |

The design pressure and the maximum and minimum pressures provide a measure of the LoS, to determine if it is adequate or excessive. The pumping head at full flow and zero flow gives the range of energy required. The reduction in energy gives a measure of the potential cost savings.

Hydraulic calculations were also undertaken for the maximum flow operating conditions, i.e., 82 hydrants at 5 L/s requiring 210 L/s delivered to a control pressure with a series of variable speed-controlled pumps (two pumps each with capacity of up to 290 $m^3$/h and one pump with capacity of up to 241 $m^3$/h).

The range of head and pressures at the hydrants around the system under maximum flow and zero flow operating conditions are shown in Table 4.

The system is designed to maintain a constant pressure and thus head at the control point in this system would be selected as Junction 14, representing the lowest elevation. Thus, at zero flow the head would be 77.66 and the total dynamic head at the pump would reduce slightly as the flow reduced.

In a pressurized pipeline with direct farmer connections, like the current system, there are many different LoS tiers that can be provided. The key LoS parameter is the pressure provided at the hydrant. From the farmers' perspective, the higher the pressure the cheaper and easier it is. On the other hand, increasing the delivery pressure results in higher supply side costs and higher water pricing. As mentioned earlier, a delivery pressure of 30 m (in the current case this

requires a pumping system head of approximately 40 m) is sufficient to provide reliable water delivery at the farm. In this system, the delivery head is much higher and could be reduced, resulting in reduced water pricing while still providing a reliable LoS.

**Table 4.** The head and pressures under various flow operating conditions.

| Node ID | Elevation (m) | Zero Flow | | Full Flow | |
|---|---|---|---|---|---|
| | | Head | Pressure | Head | Pressure |
| | | (m) | (m) | (m) | (m) |
| Junc-1 | 12.59 | 65.07 | 77.66 | 67.41 | 80.00 |
| Junc-2 | 12.30 | 65.36 | 77.66 | 67.29 | 79.59 |
| Junc-3 | 11.59 | 66.07 | 77.66 | 67.39 | 78.98 |
| Junc-4 | 11.59 | 66.07 | 77.66 | 67.00 | 78.59 |
| Junc-5 | 11.25 | 66.41 | 77.66 | 67.24 | 78.49 |
| Junc-6 | 10.29 | 67.37 | 77.66 | 68.18 | 78.47 |
| Junc-7 | 10.34 | 67.32 | 77.66 | 68.60 | 78.94 |
| Junc-8 | 11.07 | 66.59 | 77.66 | 67.51 | 78.58 |
| Junc-9 | 12.75 | 64.91 | 77.66 | 66.22 | 78.97 |
| Junc-10 | 12.59 | 65.07 | 77.66 | 65.98 | 78.57 |
| Junc-11 | 12.39 | 65.27 | 77.66 | 66.08 | 78.47 |
| Junc-12 | 11.29 | 66.37 | 77.66 | 66.81 | 78.10 |
| Junc-13 | 11.09 | 66.57 | 77.66 | 66.67 | 77.76 |
| Junc-14 | 9.21 | 68.45 | 77.66 | 68.45 | 77.66 |
| Junc-15 | 13.21 | 64.45 | 77.66 | 66.51 | 79.72 |
| Junc-16 | 12.35 | 65.31 | 77.66 | 67.15 | 79.50 |

A way of analyzing the electricity cost for water pumping is to calculate the energy required in kWh/m$^3$ using the above formula rearranged as:

$$kW = head(m) \times flow\ rate(L/s)/100/efficiency\ factor\ (0.7).$$

It is noted that 1 m$^3$ per hour equals 0.277 L/s, and the efficiency factor of 0.7 is a typical design factor that combines both the pump and electric motor performance, which can be expected in a well-designed variable speed pumping system. For a design flow of 210 L/s (758 m$^3$/h) and head of 80 m this gives an energy requirement of 240 kW. This can be expressed as 0.32 kWh/m$^3$.

If the system delivers a lower operating pressure, then the reduction in energy can also be calculated by the rearranged formula where the delivery head is 40 m not 80 m. This results in a 50% reduction from 0.32 kWh/m$^3$ to 0.16 kWh/m$^3$. In other words, for every 1 m reduction in head there will be a 0.0053 kWh/m$^3$ reduction in energy required.

The design pressure at water intake points is greater than the required pressure at spray devices (pressure at the lowest level $H_{td}$ = 65.98 m > $H_{required}$ = 30 m). High-pressure pumping leads to more electricity consumption and a high cost of electricity. If the delivery pressure is reduced to 40 m at the hydrant, then the energy saving will be approximately of 0.16 kWh/m$^3$ pumped.

*3.2. Water Price Calculations*

The water price for Thanh Nghia irrigation system is calculated using the average cost pricing method. The calculation is undertaken for scenarios that reflect assumed levels of service of irrigation water provided to the farmers.

The cost of water supply and delivery in the case of Thanh Nghia irrigation system consists of: (1) Production cost; (2) Expected profit; and (3) Taxes and fees. The production cost consists of (i) Direct production cost, (ii) Management cost, and (iii) Sales cost. The direct production cost consists of direct material cost, direct labour cost, and general production cost. These cost items are calculated based on the current Government's norms and the regulations of Thanh Nghia AC with the data collected from the Cooperative.

Direct material cost in the current case is the amount paid for consumption of electricity to operate the pumps. Different options for pump operation, with respect to given operating heads, will result in different consumptions of electricity. In addition, the electricity rate is also influenced by the time that the pumping station is operated.

It is noted that the LoS can be reflected by the time of operating the pumps. In Vietnam, three time frames of using electricity (at peak, off-peak or low-load hours) are charged at different prices (Table 5). This will influence the cost of the water supply with respect to the time of pumping.

**Table 5.** Time frames and the price of electricity by the EVN of Vietnam.

| No | Time Frame | Specific Time in a Day | Electricity Price (US$/kWh) |
|----|------------|------------------------|-----------------------------|
| 1 | Off-Peak Hours | From 04:00 to 9:30 (05 h and 30 min); From 11:30 to 17:00 (05 h and 30 min); From 20:00 to 22:00 (02 h). | 0.07 |
| 2 | Peak Hours | From 09:30 to 11:30 (02 h); From 17:00 to 20:00 (03 h). | 0.12 |
| 3 | Low-Load Hours | From 22:00 to 04:00 the next morning | 0.04 |

By consulting the time frames of water users, 98% of 100 farmers agreed with the option of water supply at off-peak hours, with only 2% agreeing with peak hours. The price of electricity in this case was calculated on the average basis of the two time frame prices given, and it is equivalent to the rate of 0.08 US$/kWh).

Various options for water prices were calculated with respect to the assumed levels of service provided to the farmers. The cost items and the water prices for Thanh Nghia irrigation system under the design conditions and the selected LoS are summarized in Table 6.

**Table 6.** Cost items and the water price by the design and selected LoS.

| No | Cost Item | Unit | Amount (USD) * | |
|----|-----------|------|----------------|--|
| | | | The Design | Selected LoS |
| 1 | Direct Material Cost | USD | 19,142.68 | 9571,34 |
| 2 | Direct Labor Cost | USD | 5267.15 | 5267.15 |
| 3 | General Production Cost | USD | 17,854.37 | 17,854.37 |
| 4 | Sales Cost | USD | 2798.35 | 2798.35 |
| 5 | Management Cost | USD | 4102.88 | 4102.88 |
| 6 | Total Production Cost (1–5) | USD | 49,165.45 | 39,594.10 |
| 7 | Expected Profit 5% (7) = (6) × 5% | USD | 2458.27 | 1979.71 |
| 8 | Total Cost (VAT Excluded) | USD | 51,623.72 | 41,573.81 |
| 9 | VAT (10%) (Not Applicable) | % | - | - |
| 10 | Total Cost (VAT Included) | USD | 51,623.72 | 41,573.81 |
| 11 | Total Quantity of Pumped Water | m$^3$ | 765,000 | 765,000 |
| 12 | **Average Cost Water Price (10/11)** | USD/m$^3$ | 0.067 | 0.054 |
| 13 | **Two-Component Water Price** | | | |
| | -Fixed Water Price per Year | USD/ha | 318.441 | 318.441 |
| | -Variable Water Price per m$^3$ | USD/m$^3$ | 0.025 | 0.013 |

Note: * The water prices were calculated with a very high LoS (the design capacity) and the LoS agreed by the farmers and Thanh Nghia AC (with the pump heads of 80 m (the design) and 40 m with water supplying at peak and off-peak time. Value Added Tax (VAT) was not applied in case of irrigation water supply. One US Dollar is equivalent to about 23,000 Vietnam Dong.

The production cost in the two cases above differs in terms of direct material cost (electricity consumption for pumping). The average cost water prices calculated for the case of the design and the selected LoS are 0.067 US$/m$^3$ and 0.054 US$/m$^3$, respectively. Meanwhile, the water prices calculated by the two-component method in the current case assumed the fixed cost was not variable and that there was only change in the cost of energy

for pumping. The fixed price by year per hectare for the case of the design and the selected LoS is the same amount of 318.441 US$/ha/year, while the variable price is 0.025 US$/m$^3$ and 0.013 US$/m$^3$ for the case of the design and the selected LoS, respectively.

The average cost water price with respect to the selected LoS has been decided for implementation with the agreement of the Management Board of the Cooperative and the farmers in a recent cooperative member meeting with an acceptance of 150 out of 150 households' representatives attending the meeting. The price level will be adjusted on the basis of a financial year when cost elements are changed, or modified with regard to changes in LoS or any actual extension of irrigated area. In these cases, the water provider and the farmers will discuss the details and agree a new water price at the annual member rally that is organized for each year.

### 3.3. Impact of Factors on Water Price

In pressurized pipeline systems, the major costs are for the electricity and the depreciation of the infrastructure, which both together are responsible for about 80% of the overall costs. Therefore, options to reduce the operating pressures (which drive electricity cost) and the overall system capacity (which drives depreciation) are very important considerations when calculating water prices. In this case study, various cases with respect to different levels of service were assumed and the water price was calculated to the system based on the cost incurred. The results show a large change in respect to the price of the irrigation service that can be selected. A number of cases relevant to the LoS were considered and it was shown how it affected the total cost and then the water price.

### 3.3.1. Reduced Pressure

The energy requirements were calculated for both the design pump pressure (80 m) and the option of reduced pump pressure (40 m). This reduction reduced the energy input by 50% and thus reduces the variable cost by 50% if only the pressure is being modified. In the current case, the energy cost represents about 37% of the total cost, therefore, the reduced pressure would result in an overall reduction of 18% of the total costs.

### 3.3.2. Time of Pumping

By consulting the farmers for their expectations with regard to the existing production and the convenience of irrigated activities, the option of a pumping time during the daytime is most favorable. However, if the farmers pumped during the off-peak period, the electricity costs would be reduced to 0.04 US$/kWh instead of 0.08 US$/kWh, and water price would reduce by 17%. However, enabling this to occur would require either: (i) an operating control by the cooperative to force farmers to only operate at that time; (ii) a labor intensive monitoring program to determine which farmers only operated at night; (iii) farmers adopt automatic control systems plus an honesty system that is strategically monitored to determine which farmers operate at night—the overall system meter could be used to reconcile the extent of compliance relatively easily and (iv) preferably the installation of individual digital meters that are time based (existing meters are not capable of this).

Farmers may install relatively cheap automatic control valves and an individual farm computer (internet connection) operated system. This would make night watering easy and acceptable. If they were installed, then option (iii) above is considered a viable option that would enable considerable cost reductions in water pricing to be achieved. The choice would be finally made by the farmers, depending on whether they may wish to invest more for their automation system to be irrigated at the night time.

### 3.3.3. Reduced Capital Costs

The depreciation is based upon the actual capital cost of the system and occupies 32.7% of the total water price cost. If the system was designed more cost effectively (that is, the system was about half of the current capacity) then the overall cost would have been

approximately halved. This would mean that the depreciation would be halved and that the overall water price would be reduced by approximately 16% overall.

### 3.3.4. Overall LoS Impact on Water Price

If the system had been designed at a lower operating pressure and a reduced capacity, then the overall water price would have been reduced significantly. In this case, the cost would have been reduced by about 34%. If the system was operated at night, then a further saving of approximately 9% would have been made. Thus, if all of these changes were made then the actual cost would be reduced by a total of 43%.

The specification of assumed LoS and agreement on the one selected help to strengthen the understanding between the water provider and the farmers, as well as to consolidate the relationship between both parties. Water pricing following this approach is appropriate for achieving its objectives, especially in pressurized pipeline irrigation systems. With an appropriate water price composed of selected LoS tiers and the commitment of the water users to pay for the service, the supply and use of the irrigation service is more likely to be efficient and sustainable.

### 3.3.5. Potential Subsidy

In the current policy framework in accordance with the Law on Hydraulic Works, a payable amount of subsidy would be paid by the government to irrigation suppliers regarding the provision of irrigation services applied at the national level. This policy is specified by Decree 96/2018/ND-CP dated 30 June 2018 of the Prime Minister on prices of irrigation and drainage products and services and subsidies to the service users. Detailed price frames applied to different regions of Vietnam are promulgated by the Ministry of Finance over time and currently the prices are regulated in Decision 1477/QD-BTC on maximum prices of irrigation and drainage products and services for the year 2021. At the provincial level, Lam Dong Provincial People's Committee also issued Document No. 5087/UBND-NN dated 8 June 2020 on the prices of irrigation services for the period 2021–2026 in the province, and the price (not including VAT) for water supply to irrigate vegetables and short-term industrial plants in the form of pumping is 40% of 1,844,000 VND/ha/season of rice production (or equivalent to about 0.009 $US/m$^3$ for the current case). When the subsidy is made on the basis of irrigating season (two seasons per year) to the Cooperative, the amount will be reimbursed in the water price in the next season.

## 4. Discussion and Conclusions

The literature in the field of irrigation has shown a variety of methods for establishing water prices in irrigated agriculture and selection of an appropriate method to apply depends on specific conditions. The application of the LoS-based water pricing method can achieve multiple objectives, through which it is able to increase the efficiency of the supply and use of water, as well as sustain irrigation services. The results from this study provide important implications for promoting irrigation policies in order to develop the irrigation sector in the future.

First, with the upgrade from an open canal irrigation system to a pressurized pipeline irrigation system, water is supplied automatically on farms using sprinkler or dipping irrigation tools. Modifying pumping pressure can also impact on the level of service that the irrigation system would provide. Therefore, we can say that modernization of irrigation systems increases service levels and farmers have more choices for services provided. This would not only help the water supplier to strengthen the management of the irrigation service in order to provide better services [14], but it would also help the water users to maximize their satisfaction regarding irrigation activities as well as agricultural production outcomes.

Evidence from the current study showed that before the modernization was undertaken, the farmers faced difficulties in getting water from the open canal systems. The water level of canals was unstable over time and the cooperative had been operating the

pumps for more hours before the irrigation time to raise the water level. Farmers had also been spent many hours per day on irrigation because these activities were undertaken manually. After the modernization was completed, the farmers' irrigation systems at the farm level were connected to the pipeline network of the cooperative irrigation system, and then irrigation activities could be undertaken automatically. The irrigation service became more secure for the farmers. When the quality of service is high and irrigation water can be achieved on demand, the productivity from agricultural production can be maximized.

Second, pricing irrigation water based on LoS means that water supply from the supplier side will be more secure, since all the costs of water supply will be considered. Critically, the results of this study show that water pricing based on LoS is more likely to be accepted by the water users. Data on the actual cost to the farmers from the current case showed that the total cost to farmers for irrigation water was only less than half in the case of modernization, compared to without modernization. This saving was mainly from the reduction in cost of equipping individual pumps and pipes to pump water from open canals to the household farms, and the reduction in labor cost for irrigation activities by more than half after the modernization was implemented. In fact, the survey results showed that all farmers asked accepted the given LoS and the determined water price accordingly. The water users' willingness to pay would be basis for sustainable irrigation infrastructure. In such situations, the long-term objective in terms of welfare embedded within the changes in water pricing over time could be achieved [28].

Third, the adoption of the LoS-based water pricing for modernized irrigation systems can be seen as a measure to address water scarcity and uncertainty in the context of climate change, as well as higher requirements of water users for agricultural production activities. In the current study, the total volume of water pumped under the new system is approximately 803,250 $m^3$ per year (irrigated for 102 ha). This compares to previous estimates of 872,302 $m^3$/year under the old system and suggests that there has been a water saving of 219,449 $m^3$ per year or 27.3% of used water quantity. This could also increase the opportunity for water use for other economic sectors such as domestic uses, industrial production and tourism development in the province, where there are many ecological tour sites involving lakes and waterways. This saved water has a very significant benefit to the system, as it will reduce the impacts of drought on other sectors, especially in the dry season. The more water that can be saved from agricultural use, the higher the economic wellbeing that could be generated for the society in general.

Nevertheless, various issues need to be seriously taken into account when this method of water pricing is applied. The technical issues of an irrigation system are important to be considered to assess the system's capability to ensure the delivery of water under different operating conditions, including the ability to measure the quantity of water used. In addition, the supplier and water users need to work closely together to agree on the most technically feasible and economical irrigation option to obtain the most reasonable water price for farmers.

Although the application of the pricing method in the current case is clear, the scaling up of this approach in practice may involve issues. First, the current case only focuses on a small-scale pressurized pipeline irrigation system. It is not difficult for the supplier and water users to understand and clarify assumed levels of service and then to agree on a selected LoS and a given water price. However, in larger scale systems with a complexity of irrigation forms, it is not easy to specify and ensure a certain LoS, as well as to create a mechanism where the supplier and water users can work closely to agree on the most feasible and efficient LoS. In this case, depending on the system, different irrigated areas with specific features can be identified to set a given LoS for pricing. Second, the concept of LoS is meaningful when the demand and willingness to pay for water is high. The LoS-based water pricing is more applicable in irrigation systems where water is used to irrigate relatively high-value crops. Third, the modernization of irrigation systems requires upgrading infrastructure. Meanwhile, financial sources are constraints, especially in the developing world. Support from the government may help to make a change toward the moderniza-

tion of irrigation systems, however, targeting and designing appropriate mechanisms for irrigation interventions are important [29]. This can also be implemented in the form of public and private partnership, which has been widely promoted in many countries.

The lessons learnt from this case study were as follows:

- The modernization of irrigation systems will secure water supply and help to promote efficient and sustainable development of irrigated agriculture. It also helps to ease the workload of farmers in agricultural production by investing and applying more modern irrigation techniques on farms.
- The choice of design overall capacity of irrigation systems is important, since the cost of fixed asset depreciation often accounts for a major part of the irrigation system.
- Water pricing with volumetric-based methods (under the average cost or two-part tariff) helps to achieve several objectives: to manage demand and discourage waste use of water resources; to ensure an appropriate prices and equitable access to water and water-related services and to balance the expenditure and revenue from water supply to cover investment and operation and maintenance costs. The two-part pricing method (including fixed and usage-based pricing) can promote efficiency while ensuring equality.
- The flexibility in choosing water pricing methods and selecting LoS would increase farmers' willingness to pay for the irrigation services. When the situation changes, the supplier and the water users can discuss how to modify and agree on a new water price. This strengthens the relationship between the supplier and the water user, and promotes the sustainability of irrigation services.
- Governments can help to solve several matters (for example: specifying a subsidized amount to different types of systems and water uses and promoting the application of the two-part water pricing method by issuing official documents for implementation).

A major obstacle to promoting the new approach of water pricing in irrigation could be changing the perception of farmers. When a cost reflective pricing is applied, expected higher prices of irrigation services may lead to the resistance of farmers, even though the actual benefit that the farmers receive surpasses the cost they have to pay. Overcoming this requires efforts to increase awareness, as well as policy reforms to further the development of the approach in practice.

**Author Contributions:** Conceptualization, methodology, T.D.T. and B.A.T.; formal analysis, B.A.T.; investigation, T.D.T. and B.A.T.; data curation, writing—original draft preparation, B.A.T.; writing—review and editing, T.D.T. All authors have read and agreed to the published version of the manuscript.

**Funding:** This research received no external funding.

**Data Availability Statement:** There are no further data available.

**Acknowledgments:** The authors would like to thank Rob Rendell from the Australian Water Partnership (AWP) for his comments and advice during the implementation of this study. The authors also express their thanks to Rebecca Schwarzman from AWP for her help in revising the language and making every sentence of the manuscript clear.

**Conflicts of Interest:** The authors declare no conflict of interest.

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
