# Peer review of "Use of Level-of-Service-Based Water Pricing to Sustain Irrigated Agriculture: A Case of Modernized Irrigation System in Vietnam"

_water, doi:10.3390/w15091780_

Round 1
Reviewer 1 Report (Previous Reviewer 3)
In my comment about two-way pricing, I was offering a compliment about the idea, not a suggestion about the wording. It was fine the way it was.
Equation 4: What are the factors of 102 and 3.6? If these are for unit conversion, they need units.
Equation 4: Gamma should be “density,” not “specific density”
Figure 2: Some parts are still illegible, misplaced, cut off, or obscured (at least in the PDF).
EPANET needs a citation, usually Rossman (2000).
Table 4: Head and pressure columns should be reversed. Head should be constant across all junctions at zero flow; pressure should vary with elevation. Then this would match the description about Junction14 immediately below the table.
The Discussion is better, but still needs some strengthening. I suggest pointing to evidences of “before and after” in this case study. How did the farmers and the water supplier perceive the LoS before the modernization of the irrigation system? And how did it change with the new infrastructure and pricing? How can that be applied beyond this individual case? What lessons were learned that can help other irrigation systems be motivated to modernize technology and pricing? Because this is a case study, such discussion is appropriate.
Author Response
2 May 2023
Dear Professor
The reviewer for the Journal of Water
We thank you very much for your insightful comments on the manuscript entitled “Use of level-of-service-based water pricing to sustain irrigated agriculture: a case of modernized irrigation system in Vietnam” submitted to the Journal of Water with the ID of [Water] Manuscript ID: water-2333068. We have carefully taken into account your individual comments and made revisions in detail to the manuscript. We hope our revisions address key points from you. We think that thanks for the comments from you, the quality of the manuscript has been much improved.
We believe that the research results presented in this manuscript would add to the literature in the field of irrigation. The water pricing method proposed in this paper is an approach that would encourage both the supplier and water users to increase their responsibility regarding water supply and service delivery so that it can promote the efficient use of water supplied with reasonable water prices, sustain the operation and management of irrigation systems, as well as increase the productivity of irrigated agriculture. We hope that the paper provides helpful lessons for policy makers and irrigation suppliers/irrigators in modifying current polices and water pricing mechanisms to promote the efficiency and sustainability of water supply for agricultural production in the future, especially in developing countries.
We thank you again for your comments and that is highly appreciated.
With best regards,
Truong Duc Toan (PhD)
Department of Economics
Thuyloi University
Hanoi, Vietnam
Bui Anh Tu (PhD)
Department of Construction Management
Thuyloi University
Hanoi, Vietnam
Comments and responses to the comments:
REVIEWER 1:
General comments:
In my comment about two-way pricing, I was offering a compliment about the idea, not a suggestion about the wording. It was fine the way it was.
Specific comments:
Comments:
Equation 4: What are the factors of 102 and 3.6? If these are for unit conversion, they need units.
Response: These factors have no unit so we do not make any changes to the manuscript.
Comments:
Equation 4: Gamma should be “density,” not “specific density”
Response: Yes, thank you for the suggestion.
Comments:
Figure 2: Some parts are still illegible, misplaced, cut off, or obscured (at least in the PDF).
Response: Some revisions were made to the comments, including moving the subsection of specification of LoS to the methodology section, adding more statements/explanations to certain parts of the manuscript.
Comments:
EPANET needs a citation, usually Rossman (2000).
Response: Thank you for your comment. We added the reference to the manuscript.
Rossman, L. A. 2000. EPANET 2 Users Manual. National Risk Management Research Laboratory. US Environmental Protection Agency, Cincinnati, OH.
Comments:
Table 4: Head and pressure columns should be reversed. Head should be constant across all junctions at zero flow; pressure should vary with elevation. Then this would match the description about Junction14 immediately below the table.
Response: Thank you for the comments. All make better sense.
Comments:
The Discussion is better, but still needs some strengthening. I suggest pointing to evidences of “before and after” in this case study. How did the farmers and the water supplier perceive the LoS before the modernization of the irrigation system? And how did it change with the new infrastructure and pricing? How can that be applied beyond this individual case? What lessons were learned that can help other irrigation systems be motivated to modernize technology and pricing? Because this is a case study, such discussion is appropriate.
Response: We have provided with more explanations to some parts in the section and added lessons learnt from the case study.

Reviewer 2 Report (Previous Reviewer 2)
The paper is significantly improved from the original version in terms of content and clarity. I have the following minor issues:
Intro
"Thus, efficient and cost reflective water pricing leads invariably to an appropriate LoS provided to the farmer at an affordable cost."
-- this statement is too strong. Soften it to reflect reality. If it was that simple, the problem would be solved.
The aim of the paper is to present an LOS-based water pricing method as a viable method -- say what it is about your new approach that makes it more viable or a viable method to promote sustainable development. What is different about your approach compared to the other approaches you present in the intro and why is that likely to lead to a better result? This should be said in the intro.
Your research approach clear in the intro, namely that you used simulations of the hydraulic performance of the system to estimate the cost of a reduced LoS and a survey of farmer acceptance of the new prices.
3. Results
-The specification of LoS is still methods and should be in the methods section.
-The EPANET simulations provide a critical input into the LoS model. More specifics should be given about the assumptions made and parameters set in this package to make the methods reproducible by another researcher. Right now, you just say "The application of this model requires the user with expe- riences in operating irrigation systems to draw appropriate simulation scenarios." -- give the parameters and settings that such a user would need to use to reproduce your result.
--Say what pressure is required to meet farmers needs and why.
--Why does the reduction in energy give a measure of the potential cost savings? The potential cost savings under what scenario?
4. Discussion & conclusions
What data supports your assertion that "Critically, the results of this study show that water pricing based on LoS is more likely to be accepted by the water users." -- Do you have data showing a lack of acceptance using other methods? Or data showing acceptance of the current method? Refer to that data specifically. I think you mean to refer to the number of farmers who accepted the lower LoS pricing - state this specifically.
Author Response
2 May 2023
Dear Professor
The reviewer for the Journal of Water
We thank you very much for your insightful comments on the manuscript entitled “Use of level-of-service-based water pricing to sustain irrigated agriculture: a case of modernized irrigation system in Vietnam” submitted to the Journal of Water with the ID of [Water] Manuscript ID: water-2333068. We have carefully taken into account your individual comments and made revisions in detail to the manuscript. We hope our revisions address key points from you. We think that thanks for the comments from you, the quality of the manuscript has been much improved.
We believe that the research results presented in this manuscript would add to the literature in the field of irrigation. The water pricing method proposed in this paper is an approach that would encourage both the supplier and water users to increase their responsibility regarding water supply and service delivery so that it can promote the efficient use of water supplied with reasonable water prices, sustain the operation and management of irrigation systems, as well as increase the productivity of irrigated agriculture. We hope that the paper provides helpful lessons for policy makers and irrigation suppliers/irrigators in modifying current polices and water pricing mechanisms to promote the efficiency and sustainability of water supply for agricultural production in the future, especially in developing countries.
We thank you again for your comments and that is highly appreciated.
With best regards,
Truong Duc Toan (PhD)
Department of Economics
Thuyloi University
Hanoi, Vietnam
Bui Anh Tu (PhD)
Department of Construction Management
Thuyloi University
Hanoi, Vietnam
Comments and responses to the comments:
REVIEWER 2:
General comments:
The paper is significantly improved from the original version in terms of content and clarity. I have the following minor issues:
Specific comments:
Comments:
Intro
"Thus, efficient and cost reflective water pricing leads invariably to an appropriate LoS provided to the farmer at an affordable cost."
-- this statement is too strong. Soften it to reflect reality. If it was that simple, the problem would be solved.
Response: Yes, we revised the statement to soften its meaning.
The sentence was revised as "Thus, efficient and cost reflective water pricing would lead to an appropriate LoS provided to the farmer at an affordable cost."
Comments:
The aim of the paper is to present an LOS-based water pricing method as a viable method -- say what it is about your new approach that makes it more viable or a viable method to promote sustainable development. What is different about your approach compared to the other approaches you present in the intro and why is that likely to lead to a better result? This should be said in the intro.
Response: We agree. We added some statements to address these aspects in the manuscript.
Comments:
Your research approach clear in the intro, namely that you used simulations of the hydraulic performance of the system to estimate the cost of a reduced LoS and a survey of farmer acceptance of the new prices.
Response: Yes, we added this statement in the introduction part to make the research approach clearer.
Comments:
- Results
-The specification of LoS is still methods and should be in the methods section.
Response: We have shifted this subsection to the methodology section.
Comments:
-The EPANET simulations provide a critical input into the LoS model. More specifics should be given about the assumptions made and parameters set in this package to make the methods reproducible by another researcher. Right now, you just say "The application of this model requires the user with expe- riences in operating irrigation systems to draw appropriate simulation scenarios." -- give the parameters and settings that such a user would need to use to reproduce your result.
Response: We think the most important thing to consider here is the experience of selecting reliable levels of delivery pressure at the farm level. In general, a delivery pressure of 30m would be sufficient to provide an efficient delivery. Therefore, hydraulic calculations are to find out the most suitable delivery pressures for the pumping system as well as for farmer expectation. We have complemented this in the manuscript.
Comments:
--Say what pressure is required to meet farmers needs and why.
--Why does the reduction in energy give a measure of the potential cost savings? The potential cost savings under what scenario?
Response: We think we have explain these in some parts of the manuscript. For example:
“… In a pressurized pipeline with direct farmer connections, like the current system, there are many different LoS that can be provided. The key LoS parameter is the pressure provided at the hydrant. From the farmers’ perspective the higher the pressure the cheaper and easier it is. On the other hand increasing the delivery pressure results in higher supply side costs and higher water pricing. As mentioned earlier, a delivery pressure of 30m (in the current case requires a pumping system head of approximate 40m) is sufficient to provide a reliable water delivery at the farm. In this system the delivery head is much higher and could be reduced resulting in reduced water pricing and still provide a reliable LoS.”
“… If the system delivered a lower operating pressure then the reduction in energy can be calculated also by the rearranged formula where the delivery head is 40m not 80m. This results in a 50% reduction from 0.32kWhrs/m3 to 0.16kWhrs/m3. In other words, for every 1m reduction in head there will be 0.0053kWhrs/m3 reduction in energy required.
The design pressure at water intake points is greater than required pressure at spray devices (pressure at the lowest level Htd = 65.98m > Hrequired = 30m). High-pressure pumping leads to more electricity consumption and a high cost of electricity. If the delivery pressure is reduced to 40m at the hydrant then the energy saving will be approximate of 0.16kWhrs/m3 pumped.”
Comments:
- Discussion & conclusions
What data supports your assertion that "Critically, the results of this study show that water pricing based on LoS is more likely to be accepted by the water users." -- Do you have data showing a lack of acceptance using other methods? Or data showing acceptance of the current method? Refer to that data specifically. I think you mean to refer to the number of farmers who accepted the lower LoS pricing - state this specifically.
Response: We thank for your comments. Yes, we have added more arguments with supported data to the assertion. In addition, some more explanations and lessons learnt were added to address the implications of the study.

Reviewer 3 Report (New Reviewer)
Comparison among the different billing techniques included may be included.
The conclusions may be improved in clarity and numbered form if allowed
Author Response
2 May 2023
Dear Professor
The reviewer for the Journal of Water
We thank you very much for your time and comments on the manuscript entitled “Use of level-of-service-based water pricing to sustain irrigated agriculture: a case of modernized irrigation system in Vietnam” submitted to the Journal of Water with the ID of [Water] Manuscript ID: water-2333068. We have looked at your comments and made some revisions in detail to the manuscript. We hope our revisions address key points from you.
We believe that the research results presented in this manuscript would add to the literature in the field of irrigation. The water pricing method proposed in this paper is an approach that would encourage both the supplier and water users to increase their responsibility regarding water supply and service delivery so that it can promote the efficient use of water supplied with reasonable water prices, sustain the operation and management of irrigation systems, as well as increase the productivity of irrigated agriculture. We hope that the paper provides helpful lessons for policy makers and irrigation suppliers/irrigators in modifying current polices and water pricing mechanisms to promote the efficiency and sustainability of water supply for agricultural production in the future, especially in developing countries.
We thank you again for your comments and that is highly appreciated.
With best regards,
Truong Duc Toan (PhD)
Department of Economics
Thuyloi University
Hanoi, Vietnam
Bui Anh Tu (PhD)
Department of Construction Management
Thuyloi University
Hanoi, Vietnam
Comments and responses to the comments:
REVIEWER 3:
Specific comments:
Comments:
Comparison among the different billing techniques included may be included.
Response: The aim of the manuscript is not focusing on billing techniques so we do not think these should be included in the manuscript.
Comments:
The conclusions may be improved in clarity and numbered form if allowed
Response: We have provided with more explanations to some parts in the section and added lessons learnt from the case study to strengthen our key findings..

This manuscript is a resubmission of an earlier submission. The following is a list of the peer review reports and author responses from that submission.
Round 1
Reviewer 1 Report
In general, the articles are of great importance because it seems to me that it is a subject that should be analyzed in all irrigation systems, not only in developing countries because water is a resource that is relevant due to its impact on society. This is based on its availability and its use, where many of the variables that were studied in the article must be incorporated.
If it seems appropriate to comment on making a small comparison of the use of water and its effect on other economic sectors other than agriculture and establishing a kind of relationship with the economic well-being of society.
The criterion of this study showing LoS-based water pricing applied to a modernized irrigation system emphasizes the importance of policies toward modernizing irrigation systems.
Reviewer 2 Report
The authors present a study that looks at level of service irrigation pricing for a specific case study of an existing irrigation scheme in Vietnam. However, in its present form, it suffers from a number of key limitations (outlined below) and does not meet the standards of Water. Therefore, I cannot recommend it for publication.
1) The novelty is minimal and unclear. The authors calculate the level of service (LoS) pricing for water offered at different times and at different pressures from an existing piped water scheme in Vietnam. However, the introduction does not clearly motivate the research objectives or novelty of the work. The authors do not adequately describe the current open challenges in water pricing that their study specifically addresses. They discuss the benefits of LoS pricing, but not the need for new LoS pricing methods or how determining the LoS pricing of this scheme will address challenges to applying LoS pricing in general. They do not discuss how to generalize their results to larger schemes or different regions, limiting the applicability of their conclusions.
2) The methods are poorly and inadequately described. e.g.,
- Two very common methods of pricing are described very generally, but only one method is actually used. The assumptions used in the cost model are not fully or clearly described - for example, the labor cost -- what salaries, numbers of works, hours worked etc is included? What is included in the 'general production cost'? Does sales cost include the labor for sales? What is the source of these costs? The authors claim they are based on 'current government's norms and the regulations of Thanh Nghia AC with the data collected from the cooperative' -- but does not specific which costs come from which sources and apply to which circumstances.
- The methods used to compute the pressures in the piped network are not described or referred to. No errors are reported on measured quantities.
- The LoS is described in terms of multiple features (Volume, Offtake, Scheduling, Reliability) - but it is not clear which features are being incorporated into the authors cost model and why those features are being selected over others.
- The methods sections is poorly written and hard to follow and some methods are given in the results.
- Surveys of farmers are referred to (e.g., L388) but it is not clear which farmers were asked questions and what they were asked.
- Electricity of 0.08US$/KWhr is used but it is not clear how this number comes from the values in Table 5.
- The Selected LoS in Table 6 is very vaguely defined and it is not clear how this LoS was selected or why. It is also not clear how this cost was specifically calculated because the language is too vague.
3) The results are not adequately described or discussed. For the reductions in cost reported, it is not clear what the initial and final system configurations are. For example, for section 3.4.1, it is not clear if only the pressure is being reduced, or if they are also applying different time of pumping assumptions. This is true of many of the cost reductions. Section 3.4.5 does not present any results from this study or discuss them, it simply describes the subsidy policy.
4) Much of the discussion (e.g., L489 - 500) is not actually based on any results from this study.
5) The conclusions described in the abstract are not backed up by data collected in this study -- e.g., the authors claim that "The results from this study show that: (i) modernization of irrigation systems increases service levels; (ii) farmers have more choices for selecting services provided; (iii) water rates can be reasonably calculated with respect to the level of irrigation services provided; and (iv) commitment between the supplier and farmers toward irrigation activities underpins the sustainability of irrigated agriculture. " -- with the exception of perhaps iii, no data is presented to support the other conclusions. With respect to iii, this is limited to the specific levels of service chosen by the authors, which is not clearly defined. They do not, for example, demonstrate a method to calculate the cost of different levels of reliability.
6) The English and grammar needs to be improved and does not yet meet the standard of Water. In some areas, it is simply not clear what the authors are trying to convey, such as L42-43, which is not a complete sentence.
Figure 2 is illegible.
I advise the authors to reconsider the novelty of the work and more clearly state this in the introduction, to more fully and clearly describe the methods and results, and to discuss the results in the context of how it responds to a broader research question.
Reviewer 3 Report
I appreciate the study and the authors’ effort to combine irrigation and level of service. This is an important nice that deserves more attention for policy and water management. I recommend major revision addressing the comments below. Specifically, the connections between the case study and the conclusions must be more thoroughly developed.
Lines 44–57: I suggest the following report (especially the executive summary) to further support these statements: “Revenue effects of water conservation and conservation pricing: issues and practices” (https://citeseerx.ist.psu.edu/document?repid=rep1&type=pdf&doi=f161fe45b860c5d69b15f79e660bba461ceb2735).
Line 61: For block/tiered pricing, I believe the earliest documented use is: “Marginal cost pricing and the new LADWP water rates” (https://escholarship.org/uc/item/936653cp).
Line 66: I agree that level-of-service-based pricing is neglected, but there are a few pertinent articles the authors should acknowledge:
• “Insights into Efficient Irrigation of Urban Landscapes: Analysis Using Remote Sensing, Parcel Data, Water Use, and Tiered Rates” (https://doi.org/10.3390/su14031427). This recent and relevant study explores irrigation in urban settings and the role of pricing in encouraging users to match their water use to the level of service. Many of the concepts overlap with the authors’ work.
• “The Next Frontier: Individualized Rates Based on Cost of Service” (https://doi.org/10.1002/awwa.1994). This is a recent one that considers how smart metering with high resolution could be leveraged to customize users’ rates. Something similar might be possible for farmers.
• “Evaluation of Customer-Driven Level of Service for Water Infrastructure Asset Management” (https://doi.org/10.1061/(ASCE)ME.1943-5479.0000293)
I am also aware of a forthcoming article on LOS rates for irrigation provided by municipal water systems. I will provide it later if I have a chance.
Lines 81–92: I like the two-way, “reflective” approach that puts pressure on both buyer and seller to find the right price for the level of service.
Figure 2 is too low resolution to read.
2.2 Case Study: The authors imply, but do not state, that they are using a model to run hydraulics. They must describe the methods/models (EPANET or similar model). When, how, and why was the model made? Is it static or EPS? How were demands allocated to its nodes? What are its limitations?
Lines 260–261: If water volume is unlimited, why is precise pricing necessary? This contradicts later conclusions (lines 502–509) about water scarcity.
Lines 320–321: Are there other operational savings besides energy?
Discussion and conclusions: Some conclusions are a stretch from the analysis. What evidence have the authors provided that modernizing irrigation systems increases service levels, that LOS-based pricing is beneficial to all parties, or that LOS-based pricing is good for droughts? I see where the authors are going and generally agree, but the connections need to be strengthened using the data in this study. As written, I see no direct support for the statements the authors have made. This part needs to be more explicitly developed.
Backmatter: The authors must complete the Autor Contributions, Funding, Data Availability Statement, Acknowledgments, and Conflicts of Interest, as applicable. Even in an initial submission they need to be filled out as they are essential for review.